# Assessment of the integrity of real-time electronic health record data used in clinical research

Jessica Liu[1], Sameer Pandya[2], Andreas Coppi[3,4], H. Patrick Young[2], Harlan M. Krumholz[3,4,5], Wade L. Schulz[2], Guannan Gong[1]*

1 Yale Cancer Center, Yale School of Medicine, New Haven, Connecticut, United States of America, 2 Department of Laboratory Medicine, Yale University School of Medicine, New Haven, Connecticut, United States of America, 3 Section of Cardiovascular Medicine, Department of Internal Medicine, Yale School of Medicine, New Haven, Connecticut, United States of America, 4 Center for Outcomes Research and Evaluation, Yale-New Haven Hospital, New Haven, Connecticut, United States of America, 5 Department of Health Policy and Management, Yale School of Public Health, New Haven, Connecticut, United States of America

* guannan.gong@yale.edu

## Abstract

### Background

Near real-time electronic health record (EHR) data offers significant potential for secondary use in research, operations, and clinical care, yet challenges remain in ensuring data quality and stability. While prior studies have assessed retrospective EHR datasets, few have systematically examined the integrity of real-time data for research readiness.

### Methods

We developed an automated benchmarking pipeline to evaluate the stability and completeness of real-time EHR data from the Yale New Haven Health clinical data warehouse, transformed into the OMOP common data model. Twenty-nine weekly snapshots of the EHR collected from July to November 2024 and twenty-two daily snapshots collected from April to May 2025 were analyzed. Benchmarks focused on (1) clinical actions such as patient additions, deletions, and merges; (2) changes in demographic variables (date of birth, gender, race, ethnicity); and (3) stability of discharge information (time and status). A synthetic dataset derived from MIMIC-III was used to validate the benchmarking code prior to large-scale analyses.

### Results

Benchmarking revealed frequent updates due to clinical actions and demographic corrections across consecutive snapshots. Demographic changes were most frequently related to race and ethnicity, highlighting potential workflow and data entry

**Data availability statement:** Data and source codes to reproduce each analysis were included in our repository (https://github.com/ComputationalHealth/patient-merge) and were provided under a permissive open-source license (MIT License).

**Funding:** The author(s) received no specific funding for this work.

**Competing interests:** Harlan Krumholz works under contract with the Centers for Medicare & Medicaid Services to support quality measurement programs; was a recipient of a research grant, through Yale, from Medtronic and the U.S. Food and Drug Administration to develop methods for post-market surveillance of medical devices; was a recipient of a research grant with Medtronic and is the recipient of a research grant from Johnson & Johnson, through Yale University, to support clinical trial data sharing; was a recipient of a research agreement, through Yale University, from the Shenzhen Center for Health Information for work to advance intelligent disease prevention and health promotion; collaborates with the National Center for Cardiovascular Diseases in Beijing; receives payment from the Arnold & Porter Law Firm for work related to the Sanofi clopidogrel litigation, from the Ben C. Martin Law Firm for work related to the Cook Celect IVC filter litigation, and from the Siegfried and Jensen Law Firm for work related to Vioxx litigation; chairs a Cardiac Scientific Advisory Board for UnitedHealth; was a participant/participant representative of the IBM Watson Health Life Sciences Board; is a member of the Advisory Board for Element Science, the Advisory Board for Facebook, and the Physician Advisory Board for Aetna; and is the co-founder of HugoHealth, a personal health information platform, and co-founder of Refactor Health, an AI-augmented data management platform for healthcare. Wade Schulz is an investigator for a research agreement, through Yale University, from the Shenzhen Center for Health Information for work to advance intelligent disease prevention and health promotion; collaborates with the National Center for Cardiovascular Diseases in Beijing; is a technical consultant to HugoHealth, a personal health information platform, and cofounder of Refactor Health, an AI-augmented data management platform for healthcare; is a consultant for Interpace Diagnostics Group, a molecular diagnostics company. Guannan Gong is the founder of CtrlTrial Inc., an AI-augmented patient screening platform for clinical trials.

inconsistencies. Discharge time and status values demonstrated instability for several days post-encounter, typically reaching a stable state within 4–7 days. These findings indicate that while near real-time EHR data provide valuable insights, the timing of data stabilization is critical for accurate secondary use.

## Conclusions

This study demonstrates the feasibility of automated benchmarking to assess the integrity of real-time EHR data and identify when such data become analysis ready. Our findings highlight key challenges for secondary use of dynamic clinical data and provide an automated framework that can be applied across health systems to support high-quality research, surveillance, and clinical trial readiness.

## Introduction

Near real-time healthcare data has the potential for broad applications beyond direct interactions between patients and clinicians. The secondary use of electronic health record (EHR) and other real-world data (RWD), such as administrative claims data, disease registries, and personal health data collected through in-home medical devices or mobile apps, has rapidly increased [1–10]. Adopting near real-time clinical data analytics can be beneficial from clinical, operational, and research perspectives – it provides the possibility to reduce costs and duplicate procedures, enable early detection of deteriorating or high-risk conditions, decrease patient waiting time, and to ensure more personalized patient treatment that enhances outcomes.

However, administrative healthcare data, such as claims data and mortality data, typically experiences lags from at least 90 days to a year or more before becoming usable for analysis in clinical research [11]. Moreover, these data may only represent a "snapshot" of patients rather than a longitudinal assessment regarding cause and effect [6]. Information extracted from the EHR has the potential to provide near real-time access to a more complete dataset than can be provided from other real-world data sources [12,13].

Still, there are notable challenges in the use of EHR data including ensuring data quality, bias detection, data access, information delivery [10,14–17] and delayed, incomplete, and erroneous data capture caused by omissions during documentation at the time of service delivery [13,18–24]. The design focus of EHR has been transactional due to its historic focus on billing [25], and the primary use in daily clinical care workflows; analytical use of real-time EHR data in clinical research is only considered a secondary use case. Previous works have proposed that consistent and standardized methods for describing, assessing, and reporting data quality (DQ) findings could aid secondary data users and consumers to better understand the potential impact of DQ on reusing data and interpreting findings. Kahn MG et al. [26,27] introduced a DQ assessment framework of EHR data from three categories– Conformance, Completeness and Plausibility: Conformance focuses strictly on the agreement of values against various technical specifications, Completeness focuses on the

absence of data of a variable, and Plausibility focuses on the reasonability or correctness of data. However, most of the studies conducted DQ assessment on retrospective EHR data sets [28–30], treating EHR data as static entities requiring retrospective quality control rather than dynamic systems requiring temporal validation. To ensure high-quality analysis and to better characterize and understand the implications of real-time EHR data and system use, three critical gaps must be addressed: (1) What kind of patient information were entered and updated? (2) How often was information updated? (3) When and how would the updated information flow into a computational platform for analysis? (4) How can we identify when near real-time EHR data has reached a "stable" status for analysis? As near real-time EHR data constantly changes during clinical workflows and is derived from data aggregation, EHR data reflects what was recorded into the systems but may not accurately reflect a patient's status.

In this study, we assessed the completeness of real-time EHR data, i.e., whether real-time EHR data has reached a stabilized stage and is ready to use for further analytics, by comparing multiple snapshots of the real-time EHR data throughout a defined timeframe. We aimed to identify the changes and consistency of EHR patient data over time. We characterized EHR data for three use cases: (1) Duplicate patient registration (2) Incorrectly documented patient demographics information (3) Incorrectly documented discharge information for discharged encounters. Our findings highlight the feasibility of applying an automated benchmarking pipeline to determine when the real-time clinical data from EHR is in analysis ready state on several use cases.

## Methods

### Overview

We conducted a retrospective study to assess the integrity of EHR data in a real-time operational environment by benchmarking 22 consecutive daily extracts and 29 weekly extracts from the Yale New Haven Health (YNHH) clinical data warehouse (Epic Caboodle). These daily snapshots were continuously transformed into the Observational Medical Outcomes Partnership (OMOP) common data model (CDM) [31] using the YNHH computational health platform (CHP), which maintains a daily-updated data pipeline with current clinical data [32]. As a DQ study based on existing and deidentified data, this work was not classified as human subjects research and did not require Institutional Review Board approval.

### Data sources

There were two types of datasets used in the study. One was a synthetic testing dataset designed to emulate the behaviors of EHR data and to validate the benchmarking code. The second set was the larger benchmarking data set extracted from the YNHH healthcare system.

We created our testing data set from the Medical Information Mart for Intensive Care III database version 1.4 (MIMIC III v1.4) for the study. MIMIC-III is a publicly available, single-center critical care database containing medical care information on 46,520 patients who were admitted between 2001–2012 to various ICUs of Beth Israel Deaconess Medical Center in Boston, Massachusetts [33]. All the MIMIC tables were transformed into the OMOP CDM through the Extract-Transform-Load (ETL) process [34].

The benchmarking data set contained daily and weekly extractions of the YNHH clinical data warehouse transformed into the OMOP CDM.

### Data analysis and statistical approaches

Source datasets were stored as parquet format files on Hadoop Distributed File System (HDFS) of the CHP Spark cluster. Data extraction and data analysis were done with custom PySpark scripts using Apache Spark (v2.3.2) [35]. Benchmarking results were stored as CSV files on HDFS of CHP. All study-specific scripts were reviewed by an independent data scientist.

We analyzed two distinct sets of EHR snapshots that differed in temporal coverage and sampling frequency. First, we used 29 weekly snapshots collected between July 30 and November 13, 2024, which represented the most complete and temporally continuous series available; these snapshots were used for the analyses presented in Figs 3 and 4. To assess whether the trends observed in 2024 were consistent later in the year, we additionally analyzed 22 daily snapshots obtained between April 12 and May 3, 2025. For the daily series, the first snapshot (April 12, 2025) served as the baseline. We verified that baseline selection did not influence results by re-running the benchmarking code using alternative baseline dates and observing comparable outputs. Analyses using daily data (22 consecutive snapshots from April 12 to May 3, 2025) are reported in the Supplementary and were used for the inpatient/outpatient BM-3 analysis, whereas all other visualized analyses rely on the weekly 2024 snapshots. Summary statistics were computed, with the median and interquartile range (IQR) of patient counts reported.

Three benchmarking assessments were performed: (1) a pre-specified analysis of database-level changes in patient records (referred to as "clinical actions") between consecutive snapshots including patient additions, deletions, merges, and demographic changes; (2) a post-hoc analysis of specific demographic information updates among patients with changes, conducted to investigate the drivers of demographic instability identified in BM-1; and (3) a pre-specified analysis of stabilization timing for discharge time and status in baseline encounters. The benchmarking framework is illustrated in Fig 1.

For Benchmark 1 (BM-1), we assessed five categories of clinical actions that occurred between two snapshots (either consecutive snapshots or all compared to baseline snapshot). These clinical actions encompass both prospective changes—such as newly added patients entering the healthcare system—and retrospective corrections, including patient ID updates, merged duplicate records, and deleted records. Benchmarking these changes helps us better understand the evolving source patient population and the data quality improvement processes in our dataset. We have used the combination of gender, DOB, race, and ethnicity to best identify the same patient in two different snapshots. This was based on two considerations: (1) Patient ID was not reliable (2) Single demographics information might be updated, but the chance of updating the whole combination was comparably low. Following the observation of frequent demographic updates in BM-1, we performed a post-hoc analysis, Benchmark 2 (BM-2), to further assessed changes in demographic information, i.e., DOB, gender, race, and ethnicity. While this information is usually considered consistent in patients, frequent changes can indicate mis-entered information or systematic data corrections. Benchmark 3 (BM-3) focused on data entries regarding discharge time and status, as the information should be collected consistently and correspondingly in clinical workflow but often is not.

### Process of preparing the testing data set

To validate the automated EHR data extraction pipeline, we simulated various clinical actions (playbook) based on the MIMIC data to generate the testing dataset. For BM-1, the clinical actions included adding new patients with different ID (AD), updating existing patients' demographics information (IR), updating existing patients' ID (IC), deleting existing patients (DL), and merging existing patients (DM). The extraction and analytical scripts caught all mocked events (sensitivity 100%). The pipeline was illustrated in Fig 2, with similar validation simulations completed for BM-2 and BM-3. Source codes to reproduce each analysis were included in our repository (https://github.com/ComputationalHealth/patient-merge) and were provided under a permissive open-source license (MIT License).

### Results

From July 30 to November 13, 2024, weekly snapshots of the person and visit_occurrence tables of the OMOP CDM were saved and used for analysis. We also analyzed 22 daily snapshots obtained between April 12 and May 3, 2025, during which the YNHH system EHR included a median of 2,403,201 unique patients and 135,271,445 encounters. Detailed statistics are listed in S1 Table.

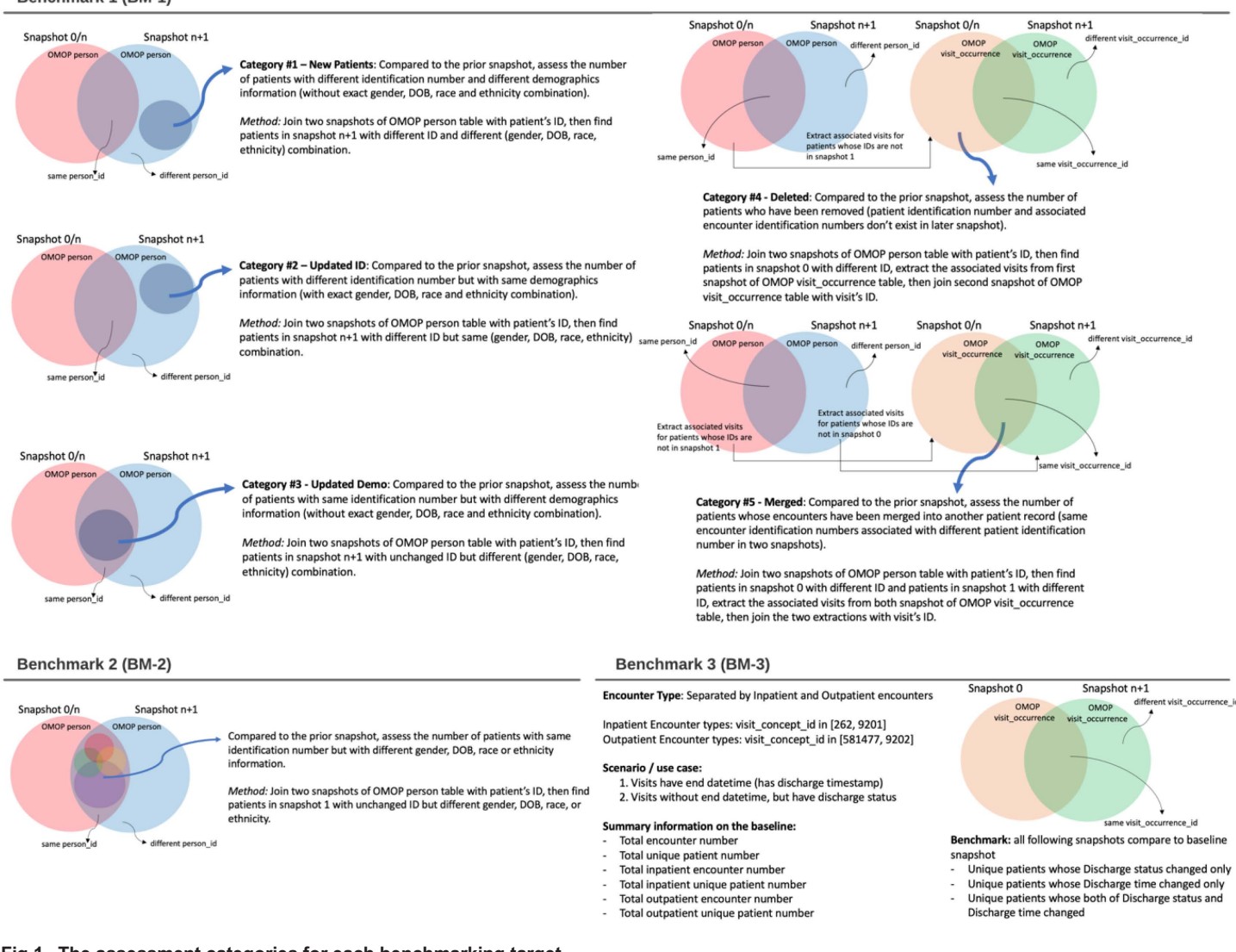

**Fig 1. The assessment categories for each benchmarking target.**

## BM-1: Categorized clinical actions detected between consecutive snapshots

The number of transactions per category were calculated for each snapshot in comparison to the previous snapshot. Between weekly snapshots from September 30 to November 13, 2024, most transactions involved new patients, followed by ID changes and demographics changes (Fig 3). This pattern was consistent between daily snapshots from April 12 to May 3, 2025, during which the median number of new patients was 817 (IQR: 332), updated patient IDs was 642 (IQR: 990), updated demographics information was 488 (IQR: 133), deleted patients was 118 (IQR: 237), and merged patient IDs was 70 (IQR: 36). The daily breakdown of number of patients in each of the clinical action categories is shown in S2 Table.

## BM-2: Categorized demographics changes detected between consecutive snapshots

The post-hoc analysis of specific demographic changes (BM-2) identified frequent updates to gender, DOB, race, and ethnicity. During the 2024 study period, these modifications were non-uniformly distributed, with peak activity occurring in

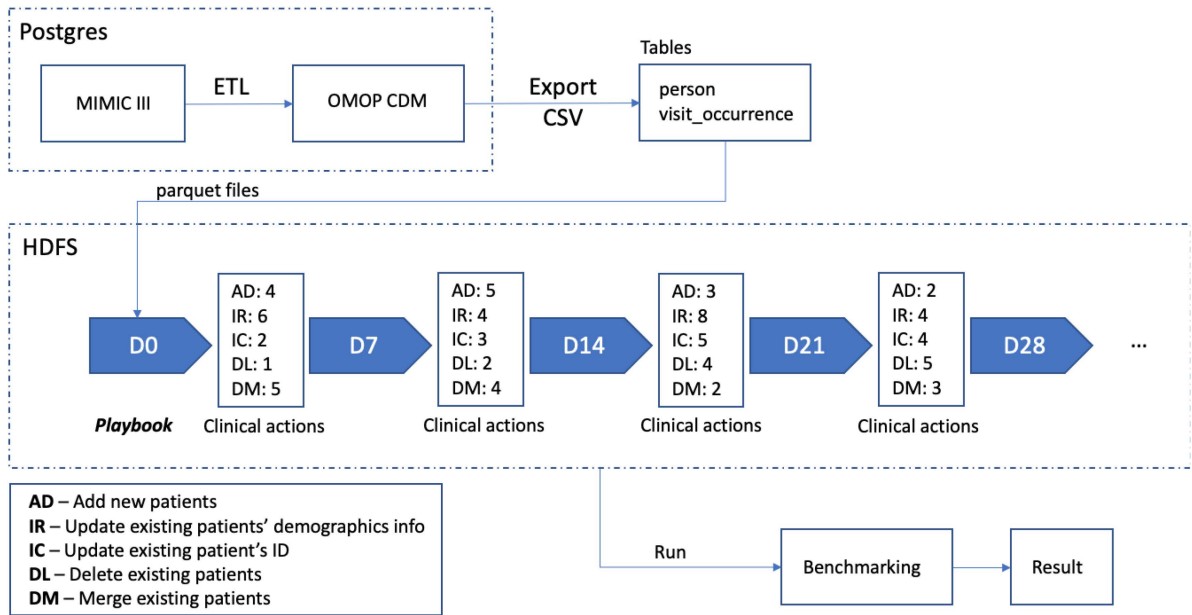

**Fig 2. Pipeline for preparing testing dataset and validating analytics code.**

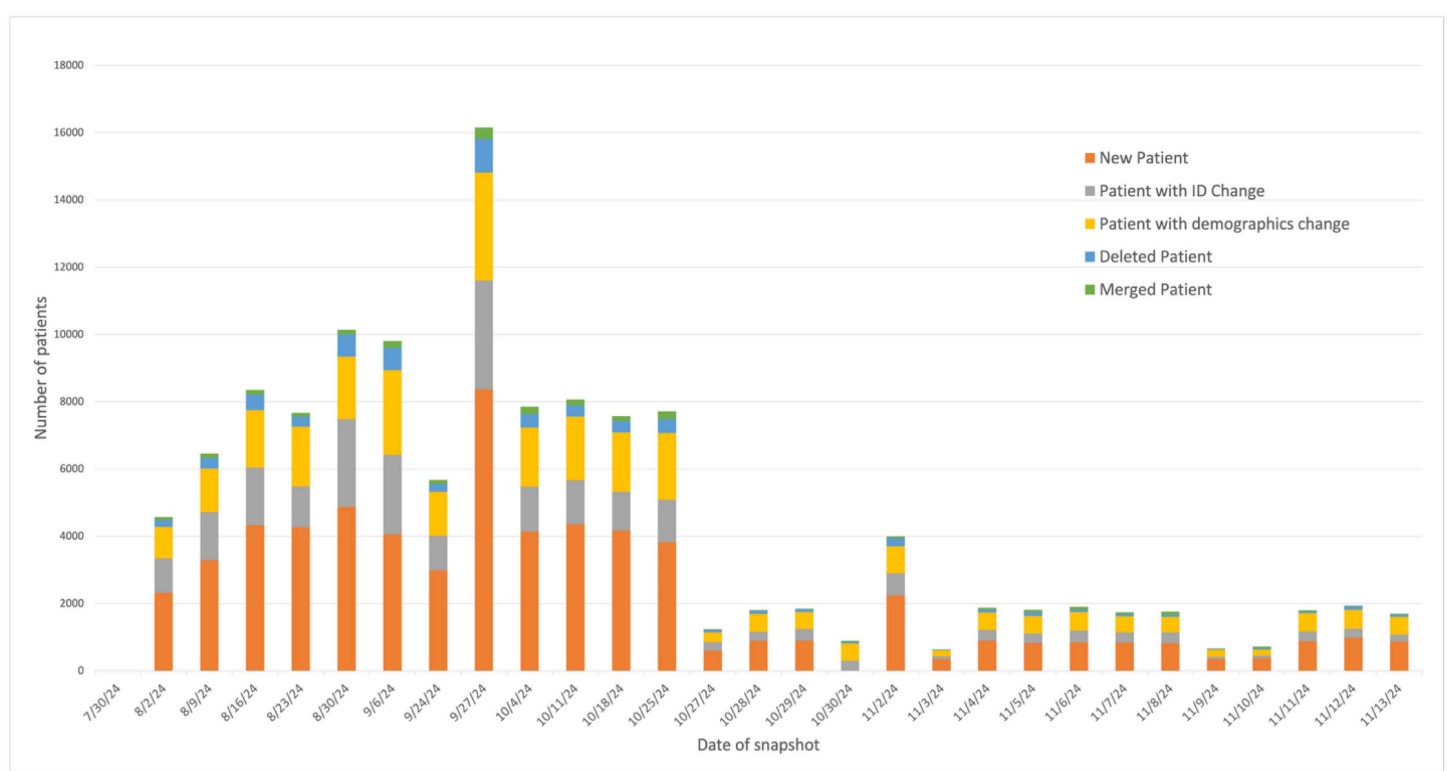

**Fig 3. Categorized EHR transactions between consecutive weekly snapshots.** Snapshots taken from July 30 to November 13, 2024.

August and October (Fig 4). The number of patients with changes in each demographics change category was calculated between each of the snapshots, i.e., comparing that day's or week's snapshot to the previous snapshot. From most to least, the median number of patients with detected EHR changes made between snapshots to race was 411 (IQR: 105), ethnicity was 119 (IQR: 48), date of birth was 15 (IQR: 19), and sex was 6 (IQR: 6). The detailed number of patients falling in each of the categories for each daily snapshot is provided in S3 Table.

**BM-3: Information updates on discharge encounters**

When patients get discharged, both discharge time and discharge status should be entered into the EHR to reflect the real-time status of patients. However, occasionally either discharge time or discharge status are updated several days after the discharge event. To assess the delay in the stable state of discharge information, we evaluated two types of patient cohorts based on their baseline snapshot data: (1) Patients with discharge time in their discharge encounter; and (2) Patients without discharge time but with discharge status in their discharge encounter. We compared continuous and consecutive daily snapshots to ensure sufficient granularity, i.e., 22 snapshots from April 12, 2025 to May 3, 2025. Assigning the first snapshot on April 12, 2025, as the baseline, we compared all the following snapshots to the baseline. On April 12, 2025, there were 994,405 patients with inpatient encounters and 1,648,863 patients with outpatient encounters in the EHR. The number of patients whose discharge status, discharge time, or both were updated was calculated for each subsequent snapshot. For all patients with inpatient encounters in the baseline snapshot, 517 patients had only their discharge status changed, 287 patients had only their discharge time changed, and 10 patients had both discharge status

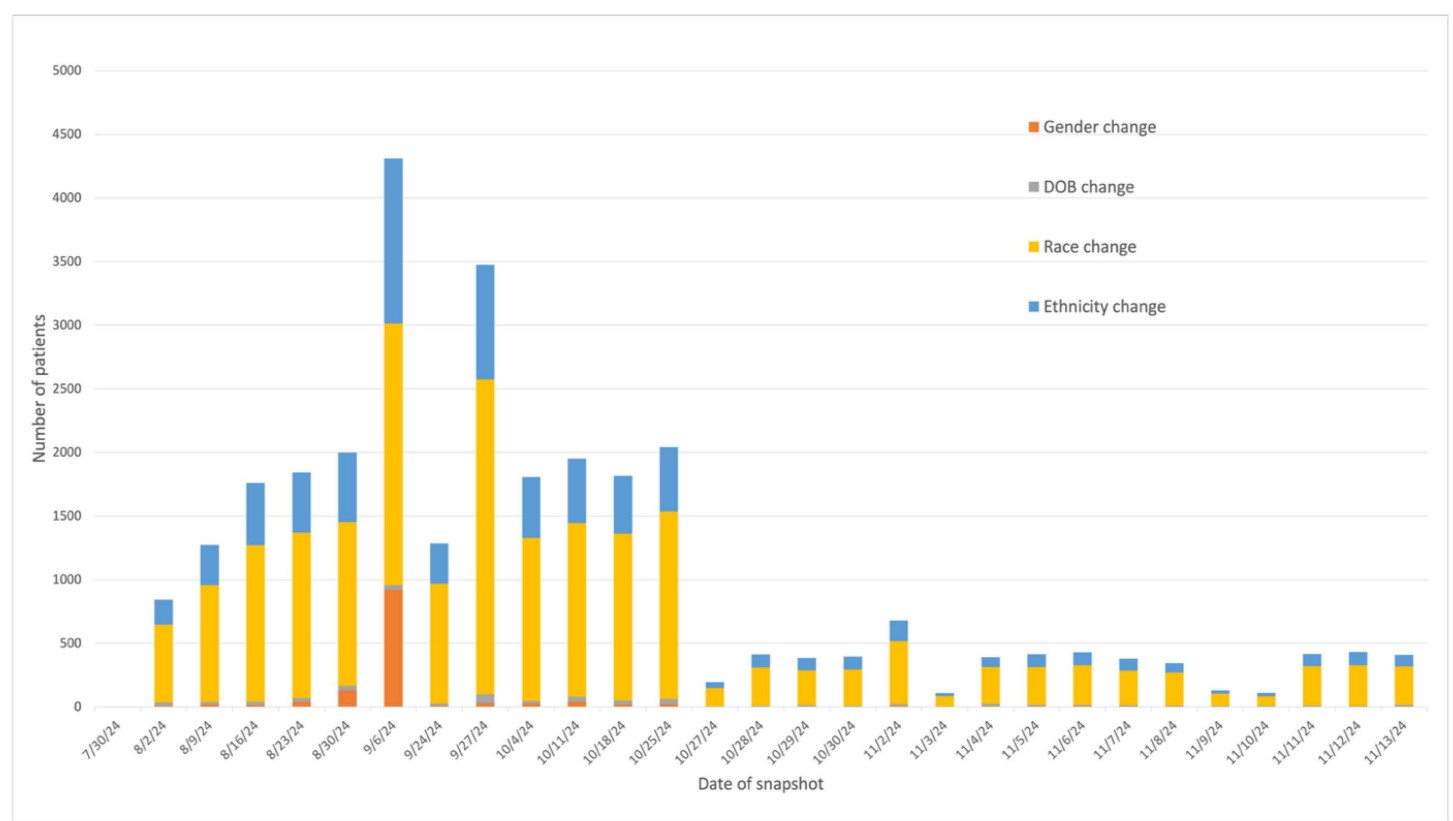

**Fig 4. Categorized demographics changes between consecutive weekly snapshots.** Snapshots taken from July 30 to November 13, 2024.

and discharge time changed by the end of the observation period (May 3, 2025). For patients with outpatient encounters in the baseline snapshot, 6,484 patients had only their discharge status changed, 996 patients had only their discharge time changed, and 144 patients had both discharge status and discharge time changed. The discharged status and discharged time information stabilized in approximately 4–7 days, though we observed a spike in discharge status changes among patients with outpatient encounters on May 2, 2025 (Fig 5). The daily counts of patients for each of the discharge change categories are provided for both inpatient (S4 Table) and outpatient cohorts (S5 Table).

## Discussion

This study systematically evaluated real-time EHR data stability and identified critical patterns that affect research readiness. Our results demonstrate three gaps that static quality assessment cannot address: First, we identified continuous clinical actions including frequent patient additions, ID changes, and demographic modifications that occur between consecutive daily and weekly snapshots (patient additions, deletions, merges) revealing that EHR data undergo constant real-time updates that static tools cannot capture. Second, demographic changes occur predominantly in race and ethnicity fields, indicating workflow inconsistencies that compromise data reliability. Race and ethnicity data at YNHH are self-assigned by patients, yet frequent updates could reflect incomplete initial collection during emergency visits that are completed in subsequent encounters, retrospective standardization to OMOP vocabulary by administrative staff, delayed patient consent after initially declining to provide this information, and reconciliation processes when patients provide different responses across YNHH facilities. Third, our discharge encounter analysis revealed that both discharge status and time required 4–7 days to stabilize post-encounter. Across the study period, more than 700 inpatient cases and 7,000 outpatient cases exhibited changes to discharge information. We also observed a sharp increase in the number of outpatient encounters with updated discharge status on May 2, 2025—from roughly 2,200–6,300 cases—which may indicate a scheduled, monthly update to discharge records. Another explanation is that certain outpatient encounters may undergo delayed finalization (e.g., completion of provider documentation, coding review, or end-of-month reconciliation workflows), leading to a large batch of discharge updates being released simultaneously. These results demonstrate that while real-time EHR data offer valuable research opportunities, data quality depends on understanding stabilization timeframes and implementing appropriate validation methods. Our benchmarking framework provides healthcare systems and researchers with evidence-based guidance for determining when dynamic clinical data become analysis-ready, addressing a critical gap in real-time clinical research.

Implementing automated real-time EHR benchmarking systems faces significant operational challenges, not only in addressing data heterogeneity and inconsistency but also in integrating teams of health IT experts, clinical informaticists, healthcare providers, and administrative staff to ensure enterprise-wide data quality and proper interpretation of observational results [21,36]. However, our benchmarking pipeline demonstrates that these challenges can be systematically addressed through validated automated approaches: we successfully used a synthetic MIMIC-III dataset to establish ground truth validation for our benchmark procedures, identified specific stabilization patterns (4–7-day periods for discharge information), and quantified demographic change frequencies that reveal workflow inconsistencies requiring targeted interventions. Therefore, while implementation complexity remains a barrier, our results provide the evidence-based framework necessary for healthcare systems to deploy automated benchmarking that delivers actionable insights to researchers and leadership, enabling informed decisions about data readiness timing and quality assurance in real-time clinical data applications.

This study has several limitations. First, as benchmarking data were collected from a single healthcare system, external validation across multiple healthcare systems would strengthen these findings. However, the temporal delays in discharge information stabilization and demographic field inconsistencies arise from how real-time Epic Caboodle data are mapped to OMOP's standardized schemas—a process common to any institution implementing OMOP CDM. Therefore, the data stability characteristics we observed are likely tied to the OMOP CDM structure and transformation process rather than

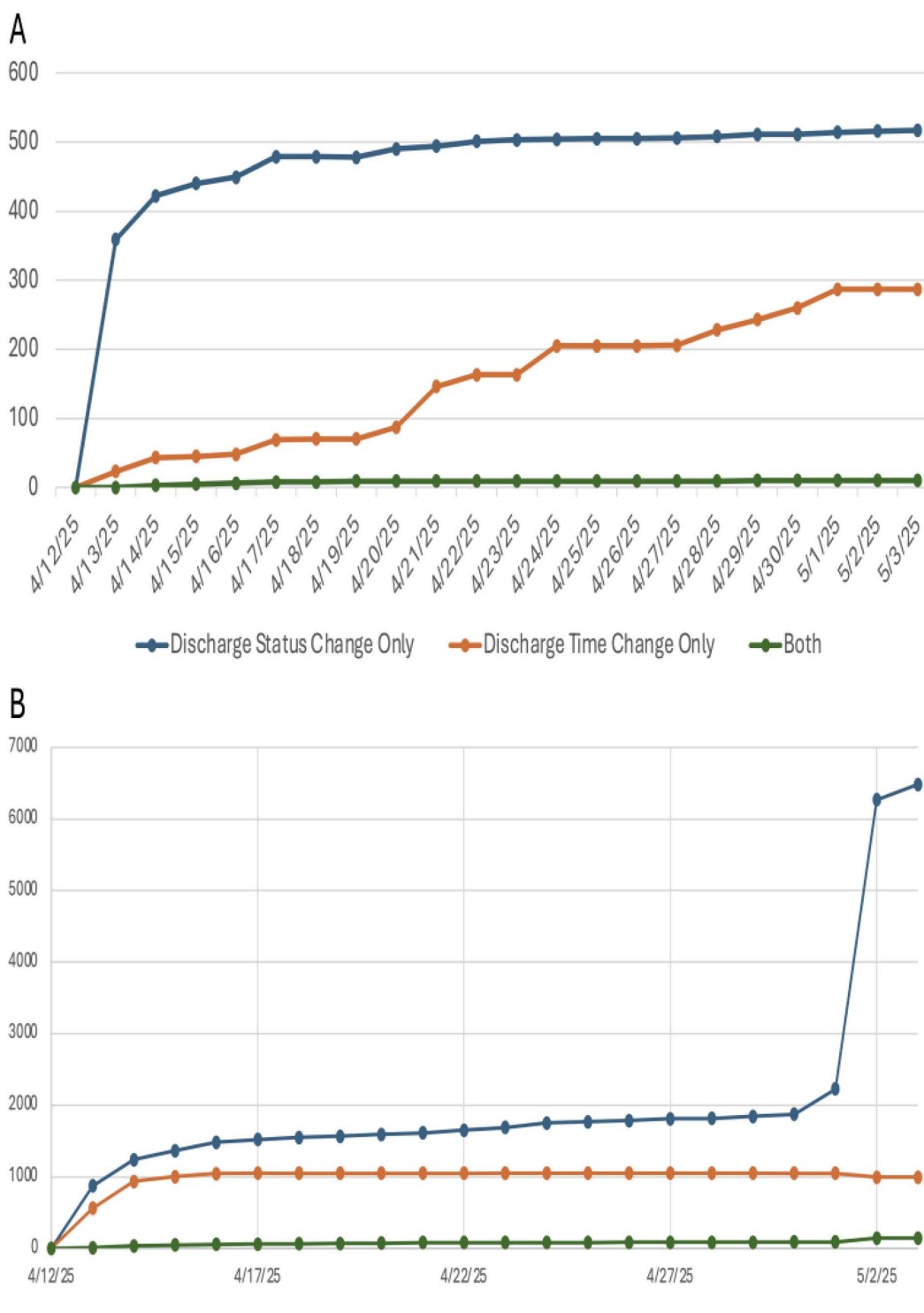

**Fig 5. The number of patients with discharge information changes, including discharge time, discharge status, or both, for patients who during the baseline snapshot on April 12, 2025, had a discharge time recorded in their discharge encounter. (A)** Patients with inpatient encounters **(B)** Patients with outpatient encounters. Discharge information was compared for each day between April 12 and May 3, 2025, with the discharge information in the baseline snapshot.

YNHH-specific system configurations. Furthermore, the pipeline in this study was designed to efficiently perform the same analysis at different healthcare systems – particularly systems with data already mapped to OMOP CDM, which has been shown to improve data quality and anomaly detection [28]. Second, the benchmarking was focused on only three use cases. Although these use cases were validated by the clinical research team as typical clinical scenarios, future work will expand this evaluation to encompass a broader range of clinical scenarios and research questions. Finally, we did not assess how to automatically integrate these insights—such as stabilization timeframes for data readiness—into researcher workflows and data aggregation processes. This integration represents an important area for future development to maximize the practical utility of our approach.

Despite the potential challenges related to the use of real-time clinical data described here, it remains a new, valuable, and rapidly evolving field. The presented approach can be used to investigate the time it takes for other EHR information, e.g., laboratory results or data on treatment responses, to stabilize and reflect the actual patient status. This strategy has been adopted in one of our recent Clinical characteristics and outcomes research for SARS-CoV-2 infection [2]. Proper assessment and validation of real-time clinical datasets will enhance the reliability and impact of research across disease surveillance, risk detection, outcome studies, and clinical trials. As healthcare systems increasingly adopt real-time data analytics, establishing standardized benchmarking approaches will be critical for ensuring data quality and supporting evidence-based clinical decision-making at scale.

## Supporting information

**S1 Table. Statistics on number of patients and associated encounters for daily snapshots between April 12 to May 31, 2025.**
(DOCX)

**S2 Table. Number of patients with EHR clinical actions taken between consecutive daily snapshots from April 12, 2025, to May 3, 2025.**
(DOCX)

**S3 Table. Number of patients with EHR demographics change performed between consecutive daily snapshots from April 12, 2025, to May 3, 2025.**
(DOCX)

**S4 Table. Number of patients with inpatient encounters whose discharge information was changed from that in the baseline snapshot of April 12, 2025.**
(DOCX)

**S5 Table.** Number of patients with outpatient encounters whose discharge information was changed from that in the baseline snapshot of April 12, 2025.
(DOCX)

## Author contributions

**Conceptualization:** Wade L Schulz, Guannan Gong.

**Data curation:** Sameer Pandya, Andreas Coppi, H. Patrick Young, Guannan Gong.

**Formal analysis:** Jessica Liu, Sameer Pandya, Andreas Coppi, Guannan Gong.

**Investigation:** Harlan M Krumholz, Wade L Schulz, Guannan Gong.

**Methodology:** Sameer Pandya, Andreas Coppi, H. Patrick Young, Guannan Gong.

**Project administration:** Harlan M Krumholz, Guannan Gong.

**Resources:** Sameer Pandya, Andreas Coppi, H. Patrick Young, Harlan M Krumholz, Wade L Schulz, Guannan Gong.

**Software:** Sameer Pandya, Andreas Coppi, H. Patrick Young, Guannan Gong.

**Supervision:** Wade L Schulz, Guannan Gong.

**Validation:** Jessica Liu, Andreas Coppi, H. Patrick Young, Guannan Gong.

**Visualization:** Guannan Gong.

**Writing – original draft:** Jessica Liu, Guannan Gong.

**Writing – review & editing:** Jessica Liu, Sameer Pandya, Andreas Coppi, H. Patrick Young, Harlan M Krumholz, Wade L Schulz, Guannan Gong.

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
