## [Decision Letter · Decision Letter 0]

24 Oct 2025

Dear Dr. Gong,

We look forward to receiving your revised manuscript.

Kind regards,

Sreeram V. Ramagopalan

Academic Editor

PLOS ONE

**Journal Requirements:**

1. When submitting your revision, we need you to address these additional requirements. Please ensure that your manuscript meets PLOS ONE's style requirements, including those for file naming. The PLOS ONE style templates can be found at https://journals.plos.org/plosone/s/file?id=wjVg/PLOSOne_formatting_sample_main_body.pdf and https://journals.plos.org/plosone/s/file?id=ba62/PLOSOne_formatting_sample_title_authors_affiliations.pdf 2. Thank you for stating the following in the Competing Interests section: Harlan Krumholz works under contract with the Centers for Medicare & Medicaid Services to support quality measurement programs; was a recipient of a research grant, through Yale, from Medtronic and the U.S. Food and Drug Administration to develop methods for post-market surveillance of medical devices; was a recipient of a research grant with Medtronic and is the recipient of a research grant from Johnson & Johnson, through Yale University, to support clinical trial data sharing; was a recipient of a research agreement, through Yale University, from the Shenzhen Center for Health Information for work to advance intelligent disease prevention and health promotion; collaborates with the National Center for Cardiovascular Diseases in Beijing; receives payment from the Arnold & Porter Law Firm for work related to the Sanofi clopidogrel litigation, from the Ben C. Martin Law Firm for work related to the Cook Celect IVC filter litigation, and from the Siegfried and Jensen Law Firm for work related to Vioxx litigation; chairs a Cardiac Scientific Advisory Board for UnitedHealth; was a participant/participant representative of the IBM Watson Health Life Sciences Board; is a member of the Advisory Board for Element Science, the Advisory Board for Facebook, and the Physician Advisory Board for Aetna; and is the co-founder of HugoHealth, a personal health information platform, and co-founder of Refactor Health, an AI-augmented data management platform for healthcare. Wade Schulz is an investigator for a research agreement, through Yale University, from the Shenzhen Center for Health Information for work to advance intelligent disease prevention and health promotion; collaborates with the National Center for Cardiovascular Diseases in Beijing; is a technical consultant to HugoHealth, a personal health information platform, and cofounder of Refactor Health, an AI-augmented data management platform for healthcare; is a consultant for Interpace Diagnostics Group, a molecular diagnostics company.Guannan Gong is the founder of CtrlTrial Inc., an AI-augmented patient screening platform for clinical trials.  We note that one or more of the authors are employed by a commercial company.  a. Please provide an amended Funding Statement declaring this commercial affiliation, as well as a statement regarding the Role of Funders in your study. If the funding organization did not play a role in the study design, data collection and analysis, decision to publish, or preparation of the manuscript and only provided financial support in the form of authors' salaries and/or research materials, please review your statements relating to the author contributions, and ensure you have specifically and accurately indicated the role(s) that these authors had in your study. You can update author roles in the Author Contributions section of the online submission form. Please also include the following statement within your amended Funding Statement. “The funder provided support in the form of salaries for authors, but did not have any additional role in the study design, data collection and analysis, decision to publish, or preparation of the manuscript. The specific roles of these authors are articulated in the ‘author contributions’ section.”If your commercial affiliation did play a role in your study, please state and explain this role within your updated Funding Statement.  b. Please also provide an updated Competing Interests Statement declaring this commercial affiliation along with any other relevant declarations relating to employment, consultancy, patents, products in development, or marketed products, etc.   Within your Competing Interests Statement, please confirm that this commercial affiliation does not alter your adherence to all PLOS ONE policies on sharing data and materials by including the following statement: "This does not alter our adherence to  PLOS ONE policies on sharing data and materials.” (as detailed online in our guide for authors http://journals.plos.org/plosone/s/competing-interests) . If this adherence statement is not accurate and  there are restrictions on sharing of data and/or materials, please state these. Please note that we cannot proceed with consideration of your article until this information has been declared. Please include both an updated Funding Statement and Competing Interests Statement in your cover letter. We will change the online submission form on your behalf. 3. Thank you for uploading your study's underlying data set. Unfortunately, the repository you have noted in your Data Availability statement does not qualify as an acceptable data repository according to PLOS's standards. At this time, please upload the minimal data set necessary to replicate your study's findings to a stable, public repository (such as figshare or Dryad) and provide us with the relevant URLs, DOIs, or accession numbers that may be used to access these data. For a list of recommended repositories and additional information on PLOS standards for data deposition, please see https://journals.plos.org/plosone/s/recommended-repositories. 4. Please include captions for your Supporting Information files at the end of your manuscript, and update any in-text citations to match accordingly. Please see our Supporting Information guidelines for more information: http://journals.plos.org/plosone/s/supporting-information. 5. If the reviewer comments include a recommendation to cite specific previously published works, please review and evaluate these publications to determine whether they are relevant and should be cited. There is no requirement to cite these works unless the editor has indicated otherwise. 

Reviewers' comments:

**Comments to the Author**

1. Is the manuscript technically sound, and do the data support the conclusions?

Reviewer #1: No

2. Has the statistical analysis been performed appropriately and rigorously?

Reviewer #1: I Don't Know

3. Have the authors made all data underlying the findings in their manuscript fully available?

Reviewer #1: No

4. Is the manuscript presented in an intelligible fashion and written in standard English?

Reviewer #1: Yes

**Reviewer #1: ** Liu and colleagues report their analysis of the integrity of real-time electronic health records (EHR) data for use in clinical research, in which they assess the stability and completeness of the recorded information in the Yale New Haven Health (YNHH) clinical data warehouse and conclude that a time lag of 4-7 days are required for records to stabilise to ensure validity for research. The work is interesting and has merit to other researchers who can utilise the framework in assessing the integrity of other local EHR systems, however I do feel the findings presented are likely to be very specific to the YNHH context, and thus the external validity of the results is uncertain.

Major comments

1. The methods section would benefit from further description of the YNHH dataset, e.g. patients that are included, sector of healthcare etc. How records enter the YNHH dataset, does each clinical encounter generate a new record, who inputs the data, etc?

2. The methods section would benefit from revision to make it clearer which samples of data used are in the analysis, currently the methods refers only to data between April/May 2025, however the axes of figures 3-5 shows data spanning July/Nov 2024, and results section also describes additional methods relating to data in November 2024. It’s not fully clear to me why the dates would change for these different assessments so this should be spelled out. It would also be helpful to know if there was any rationale for the choice of baseline dates selected and why 26 snapshots were used?

3. The results section is very descriptive and contains few statistical results, which are instead included in an appendix. I would strongly suggest adding more summary statistics to support assertions being made.

4. Conclusions “Our findings indicate that real-time EHR data require 4–7 days for discharge information stabilization and systematic monitoring of demographic field consistency to ensure research validity.” I would probably suggest revising this statement slightly to acknowledge that this is a finding within the YNHH dataset, as external validity is not proven.

Minor

5. The first benchmark assessment relates to ‘clinical actions such as additions, deletions, and merges’ – I would suggest defining exactly what ‘clinical actions’ means as it is not entirely clear to me and I’m assuming it refers to specific clinical events or healthcare encounters than have happened, and the additions, deletions or merges occur if an update has been made to either add a new event or more accurately reflect one which has taken place?

6. Race and ethnicity data were shown to be frequently updated, do you have any further insight into how race/ethnicity data are collected? Are they self-assigned or healthcare provider assigned categories? The latter tends be more prone to data quality issues therefore specifying this would be informative.

7. Careful review needed to check all acronyms spelled out at first use eg. BM-1 /consider if even needs to be abbreviated

8. Figures 3 and 4 – I am curious to understand if there is any explanations for the apparent spike in patient numbers on 09/27/24 ?

**Do you want your identity to be public for this peer review?** For information about this choice, including consent withdrawal, please see our Privacy Policy

Reviewer #1: No

---

## [Author Response · Author response to Decision Letter 1]

4 Dec 2025

Thank you for your review and comments, please see our specific response in the attached files.

---

## [Decision Letter · Decision Letter 1]

16 Dec 2025

Dear Dr. Gong,

Thank you for submitting your manuscript to PLOS ONE. After careful consideration, we feel that it has merit but does not fully meet PLOS ONE’s publication criteria as it currently stands. Therefore, we invite you to submit a revised version of the manuscript that addresses the points raised during the review process.

plosone@plos.org

We look forward to receiving your revised manuscript.

Kind regards,

Sreeram V. Ramagopalan

Academic Editor

PLOS One

**Journal Requirements:**

Reviewers' comments:

Reviewer's Responses to Questions

**Comments to the Author**

Reviewer #1: All comments have been addressed

2. Is the manuscript technically sound, and do the data support the conclusions?

Reviewer #1: Yes

3. Has the statistical analysis been performed appropriately and rigorously?

Reviewer #1: Yes

4. Have the authors made all data underlying the findings in their manuscript fully available?

Reviewer #1: Yes

5. Is the manuscript presented in an intelligible fashion and written in standard English?

Reviewer #1: Yes

Reviewer #1: Liu and colleagues have responded positively to all prior comments and the manuscript and its reporting is far strengthened by the revisions. One minor remaining comment I have relates to description of methods lingering the results section. For BM-2, it is suggested that this deeper analysis was conducted due the findings of BM-1. It would be helpful to state this earlier in the methods section, and ideally differentiate what analyses were pre-specified and which were post-hoc, which it sounds like BM-2 may have been.

**Do you want your identity to be public for this peer review?** For information about this choice, including consent withdrawal, please see our Privacy Policy

Reviewer #1: No

---

## [Author Response · Author response to Decision Letter 2]

17 Dec 2025

Per last comment, We thank the reviewer for their continued feedback and support. We have updated the Methods section to explicitly categorize BM-1 and BM-3 as pre-specified analyses, while identifying BM-2 as a post-hoc investigation sparked by the demographic instability observed in BM-1. We also relocated the methodological rationale for BM-2 from the Results to the Methods to ensure a clear separation of study design and findings. The Results section now opens directly with the quantitative findings regarding demographic updates in 2024. These revisions enhance the manuscript's clarity and adhere to the distinction between planned and exploratory analyses.

---

## [Editor Report · Decision Letter 2]

18 Dec 2025

Assessment of the Integrity of Real-Time Electronic Health Record Data used in Clinical Research

PONE-D-25-49671R2

Dear Dr. Gong,

We’re pleased to inform you that your manuscript has been judged scientifically suitable for publication and will be formally accepted for publication once it meets all outstanding technical requirements.

Kind regards,

Sreeram V. Ramagopalan

Academic Editor

PLOS One
---

## [Editor Report · Acceptance letter]

PONE-D-25-49671R2

PLOS One

Dear Dr. Gong,

I'm pleased to inform you that your manuscript has been deemed suitable for publication in PLOS One. Congratulations! Your manuscript is now being handed over to our production team.

Kind regards,

on behalf of

Dr. Sreeram V. Ramagopalan

Academic Editor

PLOS One